mathematical physics/fluid mechanics

variational calculus, Navier–Stokes equations, field description, stochastic variational description, compressible flow, sound waves

**Author for correspondence:**
M. Scholle
e-mail: markus.scholle@hs-heilbronn.de

# Competing Lagrangians for incompressible and compressible viscous flow

F. Marner[1,2], M. Scholle[2], D. Herrmann[2]
and P. H. Gaskell[1]

[1]Department of Engineering, Durham University, Durham DH1 3LE, UK
[2]Department of Mechatronics and Robotics, Heilbronn University, 74081 Heilbronn, Germany

 MS, 0000-0001-6945-5247

A recently proposed variational principle with a discontinuous Lagrangian for viscous flow is reinterpreted against the background of stochastic variational descriptions of dissipative systems, underpinning its physical basis from a different viewpoint. It is shown that additional non-classical contributions to the friction force occurring in the momentum balance vanish by time averaging. Accordingly, the discontinuous Lagrangian can alternatively be understood from the standpoint of an analogous deterministic model for irreversible processes of stochastic character. A comparison is made with established stochastic variational descriptions and an alternative deterministic approach based on a first integral of Navier–Stokes equations is undertaken. The applicability of the discontinuous Lagrangian approach for different Reynolds number regimes is discussed considering the Kolmogorov time scale. A generalization for compressible flow is elaborated and its use demonstrated for damped sound waves.

## 1. Introduction

Finding variational formulations for physical systems is beneficial with respect to a deeper understanding of the system and for establishing new solution methods, both analytical and numerical. As is well known, this methodical concept is ideally suited to, for example, the field of conservative Newtonian mechanics. Contrary to this, in continuum theories many open problems remain unsolved, typically in the field of viscous flow; since there are, in general, no obligatory construction rules for establishing variational principles, for certain problems a variety of suggestions have appeared from different authors based on different approaches. One has to distinguish between two major categories, namely between variational formulations based on a field description (Eulerian description), and stochastic variational description based on a material description (Lagrangian description) and averaging particle motion.

## 1.1. Variational formulations based on Clebsch transformation

Early attempts by Millikan [1] showed the non-existence of a Lagrangian in terms of the velocity $\boldsymbol{u}$, the pressure $p$ and their first-order derivatives that delivers the Navier–Stokes equations as Euler–Lagrange equations. A different approach based on the representation of the velocity $\boldsymbol{u}$ by three potentials, i.e. by auxiliary fields, has been established by Clebsch [2], Lamb [3], Panton [4], but in its original form his approach is restricted to inviscid flows. Later modified forms of the Clebsch transformation have been applied successfully to magnetohydrodynamics (MHD) [5] and plasma dynamics [6]. A generalized Clebsch transformation for viscous flow has been suggested by Scholle & Marner [7], but the field equations resulting from it are not self-adjoint.

Since in viscous flow dissipation[1] leads to an irreversible transfer of mechanical energy to heat, thermal degrees of freedom have to be considered in order to remain consistent with Noether's theorem which implies conservation of energy for every Lagrangian being invariant with respect to time-translations. Seliger & Whitham [8] made a first decisive step by establishing a Lagrangian that can be interpreted as a generalized form of Clebsch's Lagrangian, supplemented by two additional fields: the specific entropy $s$ and an additional field $\vartheta$, introduced three decades previously by Van Dantzig [9] as the material integral of the temperature $T$, i.e. $D_t \vartheta = T$, and termed *thermasy*, where $D_t = \partial/\partial t + \boldsymbol{u} \cdot \nabla$.

Despite including thermal degrees of freedom, Seliger and Whitham's approach remains restricted to adiabatic and therefore reversible processes. As a continuation of their work, Zuckerwar & Ash [10] suggested an extended Lagrangian considering only volume viscosity, leading to equations of motion containing qualitatively the effect of volume viscosity but differing quantitatively from the compressible Navier–Stokes equations, also known as the *Navier–Stokes–Duhem equations* [11,12] without shear viscosity. They interpret their result as a generalization of the theory of viscous flow towards thermodynamic non-equilibrium. Based on a rigorous analysis of the fundamental symmetries the Lagrangian has to fulfil, with particular regard to Galilean invariance [13], Scholle & Marner [14] suggested a Lagrangian for viscous flow considering both shear viscosity $\eta$ and volume viscosity $\eta'$ reproducing Zuckerwar and Ash's Lagrangian for the case $\eta = 0$. Again, the resulting equations of motion differ from the Navier–Stokes–Duhem equations. Considering six simple flow examples, two of them (steady shear flows) gave exact reproductions of the classical solutions, two other (transient flows) show the impact of viscosity on the flow at least in a qualitatively correct manner, whereas no physically reasonable solutions could be constructed for two pressure-driven flows.

In order to resolve the above issue, Scholle & Marner [14] made use of an analogy between quantum mechanics and fluid mechanics discovered by Madelung [15] to formulate a new Lagrangian by relating the specific entropy $s$ and the thermasy $\vartheta$ to a complex field $\chi$, termed the *field of thermal excitation* by Anthony [16], ending up with a discontinuous Lagrangian containing an additional parameter $\omega_0$. By a careful analysis it is proven that the dynamics resulting from Hamilton's principle can consistently be interpreted as a generalization of the theory of viscous flow towards thermodynamic non-equilibrium, with the parameter $\omega_0$ being the relaxation rate, giving rise to recovery of the well-known Navier–Stokes equations and the balance of inner energy when applying the limit $\omega_0 \to \infty$ to the resulting equations of motion.

## 1.2. Variational formulations based on the first integral approach

In 1994, Ranger [17] constructed an exact complex-valued first integral of the Navier–Stokes equations for steady two-dimensional (2D) flow, by introducing an auxiliary potential field $\Phi$. A first attempt to formulate a variational principle for this first integral was made by He [18,19], but is restricted to inviscid steady 2D flows only and feature unresolved issues [20]. For viscous steady 2D flows, Scholle et al. [21] made use of a Lagrangian in terms of the first-order derivatives of the streamfunction $\psi$ and the auxiliary potential field $\Phi$ in order to study inertial effects in Couette flows confined between wavy plates by means of Ritz's direct method. The first integral approach has been progressively generalized; beginning with [22] and followed by an exact complex-valued first integral of the 2D unsteady Navier–Stokes equations [23], culminating finally in the case of three-dimensional (3D) incompressible viscous flow [24], which can be derived using a potential formulation similar to that

---

[1]In the present work by *dissipation*, the irreversible transfer of energy from mechanical to thermal energy is understood from the physicist's viewpoint while the total energy (=sum of mechanical and thermal energy) is conserved. This compares strongly to the mathematician's viewpoint from which dissipation is simply understood as a loss of energy.

employed in the reduction of Maxwell's equations. By paying particular attention to the use of apposite gauge criteria for the potentials, self-adjointness can be reached at least for the steady case, leading to a variational formulation for the first integral for steady viscous 3D flow.

## 1.3. Stochastic variational description

An alternative to the above classical theoretical, deterministic field approach to the problem of finding a variational principle for viscous flow is a stochastic variational description, based on the Lagrangian equations of motion in terms of material path lines instead of a field description, making use of a statistical treatment of kinetic models (e.g. [25–28]). These start from a Lagrangian description of fluid motion and are focused on a statistical view of individual particle motion. The main idea of this approach, the roots of which are to be found in Feynman's path integrals, is that the particle motion is assumed to be stochastic with a drift that corresponds to a critical point of an action functional. Using a generalized framework provided by Arnaudon *et al.* [29] and Chen *et al.* [30], further prominent equations in continuum mechanics, e.g. the MHD equations, can be obtained.

An obvious advantage of this approach is that it is very close to classical Newtonian mechanics, where the Lagrange formalism has been successfully established, allowing adoption of many of its features.

## 1.4. Aim and scope of this paper

The variational principle formulated in [24] is based on a potential representation of the field equations and therefore of pure deterministic nature. Alternatively, scientists working on statistical physics have provided models for viscous flow based on a stochastic variational description. At first glance, these two approaches belong to disjunct concepts that have nothing in common. In this context, the work presented in [14], although originally motivated by previous research involving deterministic field theories, seems to embrace aspects of both concepts and can therefore be considered as a kind of 'in-between' of lying betwixt deterministic and statistic approach, since equations of motions result which are the classical ones plus 'deterministic' fluctuations.

The question, as to how these two different concepts compare to each other is investigated in the following. Another relevant question is how this concept applies to compressible flow.

It should be mentioned that for an arbitrary velocity field $\boldsymbol{u}$ the existence of the Clebsch variables $\Phi, \alpha, \beta$ is surely given only locally. Their global existence depends on the topological features of the flow: in the case of a non-vanishing integral of helicity, for example, for flows with closed vortex lines that form linked rings or with isolated points of zero vorticity, global existence is not a given. Further details can be found in, for example, the classical work of Balkovsky [31] and Yoshida [32] or in the more recent work of Ohkitani & Constantin [33], Cartes *et al.* [34] and Ohkitani [35]. In the case of global non-existence, completeness of the Clebsch representation requires the use of multiple pairs of variables, such as $\boldsymbol{u} = \nabla\Phi + \alpha_1\nabla\beta_1 + \alpha_2\nabla\beta_2 + \cdots$. Subsequently writing $\alpha_i\nabla\beta_i$ for $\alpha_1\nabla\beta_1 + \alpha_2\nabla\beta_2 + \cdots$ and analogously $\alpha_i D_t\beta_i$ for $\alpha_1 D_t\beta_1 + \alpha_2 D_t\beta_2 + \cdots$ aids understanding in the sense that the number of pairs has to be chosen adequately depending on the topological features of the respective individual flow problem. It is, however, not the aim of this paper to contribute to this particular research topic to which so many research groups have already contributed over several decades. In fact, the focus here is mainly on the use of discontinuous modelling of continuous systems and its statistical interpretation as well as the generalization of the concept toward compressible flow.

The paper is organized as follows: in §2 the foundations of the discontinuous Lagrangian method with time averaging are laid for incompressible flow and comparisons to alternative approaches, namely the stochastic variational description and the first integral approach, are drawn. The applicability of the different approaches for different Reynolds number regimes is discussed. The generalization of the method to compressible flow is elaborated in §3. In order to demonstrate its capabilities, the method is applied to damped acoustic waves in §4. Conclusions are drawn in §5 together with prospective further research topics.

# 2. General analysis for incompressible flow

## 2.1. The discontinuous Lagrangian for incompressible viscous flow

For incompressible flow with constant specific heat Scholle & Marner [14] suggested the Lagrangian:

$$\ell = -\varrho_0\left[D_t\Phi + \alpha_i D_t\beta_i + \frac{1}{\omega_0}\,\mathrm{Im}\,(\bar{\chi}D_t\chi) - \frac{\boldsymbol{u}^2}{2} + \bar{\chi}\chi + V - \frac{\nu}{\mathrm{i}\omega_0}\ln\sqrt{\frac{\bar{\chi}}{\chi}}\,\mathrm{tr}\,\underline{D}^2\right]. \tag{2.1}$$

Together with the velocity $\boldsymbol{u}$, it depends on the three real-valued potential fields $\Phi$, $\alpha_i$, $\beta_i$, the so-called *Clebsch variables* [2–4], and on the complex-valued field of *thermal excitation* introduced by Anthony [16]. Its absolute square delivers the inner energy[2] via

$$\bar{\chi}\chi = c_0 T, \tag{2.2}$$

where $c_0$ is the specific heat, $D_t = \partial/\partial t + \boldsymbol{u} \cdot \nabla$ the material time derivative and

$$\underline{D} = \frac{1}{2}[\nabla \otimes \boldsymbol{u} + (\nabla \otimes \boldsymbol{u})^t], \tag{2.3}$$

is the tensor of the shear rate. Here tr denotes the trace of a tensor, superscript $t$ the transpose, symbol $\otimes$ the dyadic product and a bar over a complex quantity its complex conjugate. External forces are considered via their specific potential energy $V$. Three constants appear in the above Lagrangian: the mass density $\varrho_0$, the kinematic viscosity $\nu$ and an additional constant $\omega_0$ that according to [14] can be interpreted as a relaxation rate toward thermodynamic equilibrium.

The Euler–Lagrange equations resulting from variation with respect to the elementary fields are listed below: variation with respect to the Clebsch variable $\Phi$ delivers the continuity equation,

$$\nabla \cdot \boldsymbol{u} = 0. \tag{2.4}$$

As a consequence the identity $\nabla \cdot (\xi \boldsymbol{u}) = \boldsymbol{u} \cdot \nabla \xi$, which is fulfilled for any field $\xi$, is considered subsequently. Next, variation with respect to the two remaining Clebsch variables $\alpha_i$ and $\beta_i$ lead after simple manipulation to the transport equations,

$$D_t \beta_i = 0 \tag{2.5}$$

and

$$D_t \alpha_i = 0; \tag{2.6}$$

whereas variation with respect to the velocity $\boldsymbol{u}$ delivers

$$\boldsymbol{u} + \frac{\nu}{\omega_0} \nabla \cdot \left[ \mathrm{i} \ln \sqrt{\frac{\bar{\chi}}{\chi}} 2\underline{D} \right] - \frac{1}{\omega_0} \mathrm{Im}\,(\bar{\chi}\nabla\chi) = \nabla\Phi + \alpha_i \nabla \beta_i, \tag{2.7}$$

which is a generalization of the classic Clebsch representation $\boldsymbol{u} = \nabla\Phi + \alpha_i \nabla \beta_i$ in the sense that it is not the velocity field itself that is represented by Clebsch variables but a combined vector field containing the thermal excitation, its gradient, the shear rate and its divergence.

Finally, by variation with respect to $\bar{\chi}$ the evolution equation,

$$D_t \chi + \mathrm{i}\omega_0 \chi = \frac{\nu}{2\bar{\chi}} \mathrm{tr}\,\underline{D}^2, \tag{2.8}$$

results for the thermal excitation; variation of which with respect to $\bar{\chi}$ delivers the complex conjugate of (2.8). By taking the material time derivative of (2.7) and considering the other Euler–Lagrange equations (2.4–2.6) and (2.8), gives rise to the following PDEs:

$$D_t \boldsymbol{u} = -\frac{\nabla p}{\varrho_0} + \nu \Delta \boldsymbol{u} - \nabla V + f_{\mathrm{n.e.}}, \tag{2.9}$$

as equations of motion. Equation (2.9) differs from the well-known Navier–Stokes equations by the presence of the additional forces,

$$f_{\mathrm{n.e.}} := -\frac{\nu}{\omega_0} \left\{ \mathrm{i} \ln \sqrt{\frac{\bar{\chi}}{\chi}} [\nabla \, \mathrm{tr}\,\underline{D}^2 + \{D_t + \nabla \otimes \boldsymbol{u}\}\, \Delta \boldsymbol{u}] \right.$$
$$\left. + \{D_t + \nabla \otimes \boldsymbol{u}\} \left[ 2\underline{D}\,\mathrm{Im}\left(\frac{\nabla\chi}{\chi}\right) \right] \right\}, \tag{2.10}$$

which vanish in the limit $\omega_0 \to \infty$. In the case of finite $\omega_0$, these additional terms can be classified as fluctuations due to the time evolution of the phase $\varphi = \mathrm{i} \ln \sqrt{\bar{\chi}/\chi}$ of the thermal excitation and its gradient $\mathrm{Im}(\nabla\chi/\chi) = \nabla[\mathrm{i} \ln \sqrt{\bar{\chi}/\chi}]$ which is investigated below.

A striking feature of the Lagrangian (2.1) is its discontinuity due to its dependence on the phase $\varphi$. These discontinuities become manifest along interfaces $S_n(t)$, at which several matching conditions,

---

[2]In Anthony's original work [16], the absolute square of the thermal excitation gives the temperature, but due to dimensional reasons the factor $c_0$ allows one to determine the dimension of $\chi$ as 'length per time'.

especially for the normal component of the mass flux density

$$n \cdot [[\varrho_0(u - v_s)]] = 0, \tag{2.11}$$

result from variation with respect to $\Phi$, where the double square bracket indicates the jump at an interface. $v_s$ denotes the local propagation velocity of an interface and $n$ the vector normal to it. Variation with respect to $u$ leads to the condition

$$n[[\varphi \underline{D}]] = 0, \tag{2.12}$$

indicating a jump of the shear rate vector $\underline{D}n$ at the interface due to the discontinuous phase $\varphi$, physically associated with a slip. Hence, the moving interfaces $S_n(t)$ can be interpreted as slip waves running through the fluid. In contrast, variation with respect to $\varphi$ delivers the production condition,

$$\frac{1}{2\pi} \nabla \varphi \cdot [[c_0 T(u - v_s)]] = \nu \, \mathrm{tr} \, \underline{D}^2, \tag{2.13}$$

revealing a discontinuity in the flux of the inner energy $c_0 T = \bar{\chi}\chi$ and therefore the production of inner energy due to dissipation at the interfaces. The two equations (2.8) and (2.13) together involve the irreversible transfer of kinetic energy to inner energy in accordance with the first law of thermodynamics, while the entropy increases in accordance with the second law of thermodynamics.

Independent of the question as to whether the presence of these interfaces are an artefact of the model or indeed physically interpretable as slip waves, they disappear in the limit $\omega_0 \to \infty$ like the additional forces (2.10) in the equations of motion.

## 2.2. Time averaging of the equations of motion

By considering the field equation (2.8) for the thermal excitation, one obtains the following PDE:

$$D_t \sqrt{\frac{\bar{\chi}}{\chi}} = \frac{\sqrt{\chi/\bar{\chi}} D_t \bar{\chi} - \sqrt{\bar{\chi}/\chi} D_t \chi}{2\chi}$$

$$= \frac{1}{2}\sqrt{\frac{\chi}{\bar{\chi}}}\left[ i\omega_0 \frac{\bar{\chi}}{\chi} + \frac{\nu}{2\chi^2} \, \mathrm{tr} \, \underline{D}^2 \right] - \frac{1}{2}\sqrt{\frac{\bar{\chi}}{\chi}}\left[ -i\omega_0 + \frac{\nu}{2\bar{\chi}\chi} \, \mathrm{tr} \, \underline{D}^2 \right] = i\omega_0 \sqrt{\frac{\bar{\chi}}{\chi}},$$

the corresponding general solution of which being $\sqrt{\bar{\chi}/\chi} = \exp(i\omega_0 t - i\varphi_0)$, where $D_t \varphi_0 = 0$. This implies

$$i \ln \sqrt{\frac{\bar{\chi}}{\chi}} = S(\varphi_0 - \omega_0 t), \tag{2.14}$$

with sawtooth function $S(x) = x - 2\pi \lfloor (x + \pi)/2\pi \rfloor$. The fluctuating character of the term $i \ln \sqrt{\bar{\chi}/\chi}$ is thus identified, although these *fluctuations* have to be understood in a pure deterministic sense as rapid sawtooth-shaped oscillations.

Next, by introduction of the time averaging operation

$$\langle \psi(x, t) \rangle := \frac{1}{2\tau} \int_{t-\tau}^{t+\tau} \psi(x, t') \, dt', \tag{2.15}$$

any discontinuity is eliminated from the equations of motion. Since the value of an integral over a sawtooth function is generally limited by

$$-\frac{\pi^2}{8} \leq \int_a^b S(x) \, dx \leq \frac{\pi^2}{8},$$

for any choice of the interval limits $a$ and $b$, expression (2.14), when time averaged, gives

$$\left\langle i \ln \sqrt{\frac{\bar{\chi}}{\chi}} \right\rangle = \frac{1}{2\tau} \int_{t-\tau}^{t+\tau} S(\varphi_0 - \omega_0 t') \, dt' = \frac{1}{2\omega_0 \tau} \int_{\omega_0[t-\tau]}^{\omega_0[t+\tau]} S(\varphi_0 - \xi) \, d\xi, \tag{2.16}$$

and therefore the following restriction on its absolute value:

$$\left| \left\langle i \ln \sqrt{\frac{\bar{\chi}}{\chi}} \right\rangle \right| \leq \frac{\pi^2}{16\omega_0 \tau}.$$

Furthermore it is assumed that the averaging time span $\tau$ is much shorter than the typical time scale over which temporal changes occur in the macroscopic flow. As a consequence, time averaging has no relevant effect on the remaining terms in (2.10); thus after time averaging, the additional forces fulfil

the condition

$$\langle f_{\mathrm{n.e.}} \rangle = \mathcal{O}\left(\omega_0^{-2}\tau^{-1}\right); \qquad (2.17)$$

showing that apart from regularization by time averaging the fluctuating forces are effectively eliminated from the equations of motion. On the one hand they now tend to zero quadratically with respect to $\omega_0$, while on the other they are also proportionally reduced by increasing the averaging time span $\tau$.

## 2.3. Comparison with the stochastic variational description

Despite the many differences between both approaches, a common feature of the two is that the Navier–Stokes equations are obtained by time averaging. In the case of a stochastic variational description time averaging is applied to particle motion, whereas in the discontinuous field approach time averaging is, according to (2.16), applied to the phase of the complex field of thermal excitation, indicating deviation from the local thermal equilibrium. In this context, the rapid temporal oscillations of $i \ln \sqrt{\bar{\chi}/\chi}$ accommodates the stochastic nature of the microscopic particle movement in the sense of an analogous deterministic model.

In contrast to the stochastic variational description the discontinuous field approach may alternatively be used with spatial averaging instead of time averaging. A detailed analysis of this variant of the theory is under construction and will be provided in forthcoming articles. Considering the ergodic hypothesis it is expected that, similar to turbulence theory, spatial averaging leads to the same result as time averaging.

In order to acquire a deeper understanding of the physical system, both approaches are of value. However, for establishing new solution methods, analytical as well as numerical, the deterministic field approach suggested by Scholle & Marner [14] may be more advantageous, especially for flows in finite domains, where no-slip/no-penetration conditions have to be fulfilled at solid boundaries. Using a field approach, the formulation of these conditions is straight forward; conversely, they are challenging when using a stochastic-Lagrangian approach [36].

## 2.4. Comparison with the first integral approach

A decisive feature of the Lagrangian (2.1) is the use of potential fields, which is obligatory as shown in [1,13]. The question arises, whether the Clebsch transformation is the only potential representation transforming the equations of motion into a self-adjoint form. At least for steady incompressible flow an alternative approach has been established recently by Scholle *et al.* [24], suggesting the Lagrangian in tensor notation

$$\ell = \varrho \bar{a}_{ij} u_i u_j + \left[2\eta u_j - \frac{1}{3}\partial_j \Phi\right]\partial_i \bar{a}_{ij} + \frac{1}{2}\varepsilon_{ilk}\varepsilon_{jpq}\partial_l \bar{a}_{ij}\partial_p \bar{a}_{kq} + \eta^2 u_i^2 + \frac{1}{12}\left(\partial_i \Phi\right)^2, \qquad (2.18)$$

where the velocity field is, according to $u_i = \varepsilon_{inm}\partial_n \Psi_m$, expressed by a streamfunction vector $\Psi_m$, which is quite different from the Clebsch representation; $\varepsilon_{inm}$ denotes the Levi-Civita symbol and Einstein's summation convention is used. Apart from this, a scalar potential $\Phi$ and a traceless symmetric tensor potential $\bar{a}_{ij}$ enter the above Lagrangian. Variation of the action integral

$$\delta \iiint_V \ell(\bar{a}_{kq}, a_{nn}, \partial_n \Psi_m, \partial_p \bar{a}_{kq}, \partial_i a_{nn})\,\mathrm{d}V = 0, \qquad (2.19)$$

with respect to $\bar{a}_{ij}$ results in

$$\begin{aligned}
\varrho\left[u_i u_j - \frac{u_k u_k}{3}\delta_{ij}\right] &- \eta[\partial_i u_j + \partial_j u_i] \\
&= \left[\varepsilon_{ilk}\varepsilon_{jpq} - \varepsilon_{nlk}\varepsilon_{npq}\frac{\delta_{ij}}{3}\right]\partial_l \partial_p \bar{a}_{kq} + \frac{1}{3}[\partial_l \partial_l \Phi \delta_{ij} - \partial_i \partial_j \Phi],
\end{aligned} \qquad (2.20)$$

which is a *first integral of steady Navier–Stokes equations* in the sense that taking the divergence of this tensor equation delivers the steady Navier–Stokes equations. Variation with respect to $\Psi_m$ and $\Phi$ lead to gauge conditions for the potentials; for further details see [24].

In contrast to the Lagrangian (2.1), based on the Clebsch transformation, that of (2.18), based on the first integral approach, results in a continuous Lagrangian. While applying properly to steady incompressible flows, a generalization to unsteady and compressible 3D viscous flow has hitherto remained elusive.

## 2.5. Applicability of the approaches for different flow regimes

The character of a flow is mainly dominated by its Reynolds number, being a measure for the ratio of inertial to viscous forces. The major two regimes are the laminar one, from vanishingly small up to moderate values, and the turbulent one, from its onset at a critical Reynolds number depending on the flow geometry up to very high values.

For fully developed turbulent flow, Kolmogorov [37] identified a typical time scale indicated by the so-called Kolmogorov time $\tau_\eta = \sqrt{\nu/\epsilon}$, where $\epsilon$ is the specific dissipation rate. At very high Reynolds number the Kolmogorov time may fall below the non-equilibrium relaxation time, i.e. $2\pi/\tau_\eta > \omega_0$. In this case the time averaging as performed in §2.2 can be omitted and the discontinuous approach based on the Lagrangian (2.1) can be applied since the discontinuities it includes are without relevance and the non-equilibrium restoration force caused by the latter is small compared to the classical terms in the equation of motion.

For lower Reynolds numbers, the influence of the discontinuous term in the Lagrangian (2.1) becomes increasingly relevant. Especially for small and vanishingly small Reynolds numbers and therefore laminar flow, the Lagrangian given by (2.18) based on the first integral approach, §2.4, may prove a good alternative, provided that the flow is steady.

A stochastic-Lagrangian approach is principally applicable to flows at arbitrary Reynolds numbers, but remains challenging with respect to boundary conditions along solid boundaries.

# 3. Generalization toward compressible flows

It is demonstrated that the approach is not restricted to incompressible flows; it applies equally well to compressible flow in a quite analogous way, as shown below.

## 3.1. The discontinuous Lagrangian for compressible viscous flow

Although the focus of article [14] was mainly on incompressible flow, it provides a clear hint apropos generalization to compressible viscous flow by proposing a Lagrangian, but without computing the resulting equations of motion. The following Lagrangian:

$$\ell = -\varrho\left[D_t\Phi + \alpha_i D_t\beta_i + \frac{1}{\omega_0}\,\mathrm{Im}\,(\bar{\chi}D_t\chi) - \frac{u^2}{2} + e + V\right] + \frac{1}{\mathrm{i}\omega_0}\ln\sqrt{\frac{\bar{\chi}}{\chi}}\left[\eta\,\mathrm{tr}\,\underline{D}^2 + \frac{\eta'}{2}(\nabla\cdot u)^2\right] \qquad (3.1)$$

is the one suggested in [14] after slight modifications,[3] containing the additional contribution $\eta'(\nabla\cdot u)^2$ to the dissipation rate with bulk viscosity $\eta'$. In contrast to the incompressible theory the density $\varrho$ appears as an independent field; the tensor of the shear rate is again given by (2.3) and $V$ denotes again the specific potential energy of external forces. Also for compressible flow the thermodynamics become more relevant: first the field of thermal excitation has a slightly different meaning; its absolute square results in

$$\bar{\chi}\chi = c_0 T_0\exp\left(\frac{s}{c_0}\right), \qquad (3.2)$$

being a generalization of (2.2) with specific entropy $s$. By $e$ the specific inner energy is denoted, which in classical thermodynamics is a function $s$ and $\varrho$, fulfilling the constitutive relations $\partial e/\partial s = T$ and $\varrho^2\partial e/\partial\varrho = p$ for the temperature and the pressure, respectively. In (3.1) the specific entropy $s$ has, according to (3.2), to be expressed in terms of the thermal excitation as $s = c_0\ln(\bar{\chi}\chi/c_0 T_0)$, implying

$$\frac{\partial e}{\partial\chi} = \underbrace{\frac{\partial e}{\partial s}}_{T}\frac{\partial s}{\partial\chi} = \frac{c_0 T}{\chi} \qquad (3.3)$$

and the respective complex conjugate relation.

Overall the Lagrangian (3.1) is based on the following independent fields: the velocity $u$, the mass density $\varrho$, the Clebsch variables $\Phi$, $\alpha_i$, $\beta_i$, the complex field of thermal excitation $\chi$ and its complex conjugate.

---

[3]The dissipative term $(\bar{\chi}\chi/\mathrm{i}\omega_0 c_0 T)\ln\sqrt{\bar{\chi}/\chi}[\cdots]$ is simplified to the form $(1/\mathrm{i}\omega_0)\ln\sqrt{\bar{\chi}/\chi}[\cdots]$. This has minor consequences for the equations of motion, but simplifies their derivation relevantly. The second modification is the additional specific potential energy $V$ of external forces.

## 3.2. Euler–Lagrange equations

The associated Euler–Lagrange equations are now computed. First, variation with respect to the Clebsch variable $\Phi$ delivers the continuity equation,

$$\partial_t \varrho + \nabla \cdot (\varrho \boldsymbol{u}) = 0. \tag{3.4}$$

As a consequence the identity $\partial_t(\varrho\xi) + \nabla \cdot (\varrho\xi\boldsymbol{u}) = D_t\xi$, which is fulfilled for any field $\xi$, is considered subsequently. Next, variation with respect to the two remaining Clebsch variables $\alpha_i$ and $\beta_i$ lead, after simple manipulation, to the transport equations,

$$D_t\beta_i = 0 \tag{3.5}$$

and

$$D_t\alpha_i = 0. \tag{3.6}$$

By variation with respect to the velocity $\boldsymbol{u}$ the equation

$$\varrho\boldsymbol{u} + \frac{1}{\omega_0}\nabla \cdot \left(\mathrm{i}\ln\sqrt{\frac{\bar{\chi}}{\chi}}[2\eta\underline{D} + \eta'\nabla \cdot \boldsymbol{u}\,\underline{1}]\right) = \varrho\left[\nabla\Phi + \alpha_i\nabla\beta_i + \frac{1}{\omega_0}\,\mathrm{Im}(\bar{\chi}\nabla\chi)\right], \tag{3.7}$$

is obtained, whereas variation with respect to $\varrho$ delivers

$$D_t\Phi + \alpha_i D_t\beta_i + \frac{1}{\omega_0}\,\mathrm{Im}(\bar{\chi}D_t\chi) - \frac{\boldsymbol{u}^2}{2} + e + V = -\varrho\frac{\partial e}{\partial\varrho}. \tag{3.8}$$

Finally, the Euler–Lagrange equation related to variation with respect to $\bar{\chi}$ leads to the evolution equation,

$$D_t\chi + \mathrm{i}\omega_0\frac{c_0 T}{\bar{\chi}} = \frac{1}{2\varrho\bar{\chi}}\left[\eta\,\mathrm{tr}\,\underline{D}^2 + \frac{\eta'}{2}(\nabla \cdot \boldsymbol{u})^2\right], \tag{3.9}$$

for the thermal excitation; variation of which with respect to $\bar{\chi}$ delivers the complex conjugate of (3.9).

## 3.3. Equations of motion

In appendix A the evolution equation (A 2),

$$D_t\boldsymbol{u} = -\frac{\nabla p}{\varrho} - \nabla V + \frac{\eta}{\varrho}\Delta\boldsymbol{u} + \left(\eta + \frac{\eta'}{\varrho}\right)\nabla(\nabla \cdot \boldsymbol{u}) + \frac{1}{\varrho}\boldsymbol{f}_{\mathrm{n.e.}}, \tag{3.10}$$

is derived from the Euler–Lagrange equations, and is obviously the Navier–Stokes–Duhem equation for constant viscosities with additional fluctuating forces $\boldsymbol{f}_{\mathrm{n.e.}}$ given by (A 3). As in the incompressible case, the fluctuations vanish linearly with increasing relaxation rate $\omega_0$ according to $\boldsymbol{f}_{\mathrm{n.e.}} = \mathcal{O}(\omega_0^{-1})$.

Again the convergence can be improved to $\langle\boldsymbol{f}_{\mathrm{n.e.}}\rangle = \mathcal{O}(\omega_0^{-2}\tau^{-1})$ by time averaging as in §2.2, as exemplified for damped acoustic waves below and in appendix B.

# 4. Damped acoustic waves

Sound waves can be obtained as solutions of the linearized fluid equations of motion. Much research has been done on this topic, see e.g. the review article of Jordan [38]. If transmitted through a fluid medium over a long distance, damping due to dissipation may become a relevant effect. Two competing mechanisms for damping exist, based on thermal conductivity and on viscosity. For some special fluids, like for example pure water, the thermal conductivity can be neglected [39]. This model assumption is adopted henceforth.

## 4.1. Wave geometry

Planar waves associated with the flow geometry are considered,

$$\boldsymbol{u} = u(x, t)\boldsymbol{e}_x \tag{4.1}$$

and

$$p = p_0 + P(x, t), \tag{4.2}$$

where $p_0$ is the ambient pressure. As a consequence, the friction tensor and the friction force read

$$\underline{T}_{\mathrm{v}} = [2\eta\underline{D} + \eta'\nabla \cdot \boldsymbol{u}\,\underline{1}] = \bar{\eta}\partial_x u\boldsymbol{e}_x \otimes \boldsymbol{e}_x + \eta'\partial_x u[\boldsymbol{e}_y \otimes \boldsymbol{e}_y + \boldsymbol{e}_z \otimes \boldsymbol{e}_z] \tag{4.3}$$

and

$$f_v = \eta\Delta u + (\eta + \eta')\nabla(\nabla \cdot u) = \bar{\eta}\partial_x^2 u e_x,$$ (4.4)

where the abbreviation $\bar{\eta} := 2\eta + \eta'$ is used.

## 4.2. Linearized equations of motion

Assuming $u$ and $P$ as small perturbations, quadratic and higher order terms can be neglected throughout the entire set of equations. Since the entropy production is quadratic, the process is nearly adiabatic, allowing one to assume a barotropic relation of the form

$$\varrho = \varrho(p) = \varrho(p_0 + P) \approx \underbrace{\varrho(p_0)}_{=:\varrho_0} + \underbrace{\frac{d\varrho}{dp}\Big|_{p=p_0}}_{=:a_0^{-2}} P = \varrho_0 + \frac{P}{a_0^2},$$ (4.5)

with $a_0$ being the speed of sound of undamped waves. The continuity equation then takes the linearized form

$$0 \approx \frac{\partial_t P}{a_0^2} + u\frac{\partial_x P}{a_0^2} + \left[\varrho_0 + \frac{P}{a_0^2}\right]\partial_x u \approx \frac{\partial_t P}{a_0^2} + \varrho_0\partial_x u.$$ (4.6)

Regarding the evolution equation (A 1) for the phase of the thermal excitation and considering $\omega \approx \omega_0$, the same solution (2.14) is achieved as in the incompressible case

$$i\ln\sqrt{\frac{\bar{\chi}}{\chi}} = S(\varphi_0 - \omega_0 t),$$ (4.7)

where $\varphi_0$ has to fulfil $D_t\varphi_0 = 0$, which after linearization simplifies to $\partial_t\varphi_0 = 0$ and therefore implies $\varphi_0 = \varphi_0(x)$. Based on the above findings, the non-equilibrium forces (A 3) simplify via linearization to

$$\begin{aligned}f_{n.e.} &\approx -\frac{\varrho}{\omega_0}\left[S(\varphi_0(x) - \omega_0 t)\partial_t\left(\frac{f_v}{\varrho}\right) + \partial_t\left(\frac{T_v}{\varrho}\nabla\varphi_0(x)\right)\right]\\ &\approx -\frac{\bar{\eta}}{\omega_0}[S(\varphi_0(x) - \omega_0 t)\partial_t\partial_x^2 u + \varphi_0'(x)\partial_t\partial_x u]e_x\end{aligned}$$ (4.8)

Finally, neglecting the influence of any external force ($V = 0$), the $x$-component of the equation of motion (3.10) as a linear approximation reads

$$\partial_t u = -\frac{\partial_x P}{\varrho_0} + \frac{\bar{\eta}}{\varrho_0}\partial_x^2 u - \frac{\bar{\eta}}{\varrho_0\omega_0}[S(\varphi_0(x) - \omega_0 t)\partial_t\partial_x^2 u + \varphi_0'(x)\partial_t\partial_x u],$$ (4.9)

while its $y$- and $z$-component vanish identically. Thus, via the two PDEs (4.6, 4.9) the time evolution of the velocity $u$ and the acoustic pressure $P$ is determined. The initial phase $\varphi_0(x)$ is still arbitrary; however, if homogeneity of the medium is assumed, it should be a constant. Without loss of generality $\varphi_0 = 0$ is assumed from here on.

By elimination of the velocity the two PDEs can be reduced to one PDE of higher order: by taking the derivative of (4.9) with respect to $x$,

$$\varrho_0\partial_t\partial_x u + \partial_x^2 P = \bar{\eta}\partial_x^3 u - \frac{\bar{\eta}}{\omega_0}S(-\omega_0 t)\partial_t\partial_x^3 u,$$ (4.10)

and using (4.6) in order to replace $\partial_x u$ by $-\partial_t P/(\varrho a_0^2)$, leading finally to a fourth order PDE,

$$\Box P = \frac{\bar{\eta}}{\varrho_0 a_0^2}\partial_x^2\partial_t P - \frac{\bar{\eta}}{\varrho_0\omega_0 a_0^2}S(-\omega_0 t)\partial_t^2\partial_x^2 P,$$ (4.11)

for the acoustic pressure, where $\Box$ denotes the d'Alembert operator,

$$\Box = \frac{1}{a_0^2}\partial_t^2 - \partial_x^2.$$

For $\bar{\eta} = 0$ the d'Alembert equation is simply $\Box P = 0$, the solution of which is undamped harmonic waves without dispersion.

## 4.3. Solution for the classical limit

In the limit case $\omega_0 \to \infty$ the following equation is obtained:

$$\Box P = \frac{\bar{\eta}}{\varrho_0 a_0^2} \partial_x^2 \partial_t P, \tag{4.12}$$

resulting from the classical Navier–Stokes theory. This equation has wave-like solutions of the harmonic form $P = \hat{P} \exp(\mathrm{i}[kx - \omega_1 t])$, provided that $k$ and $\omega_1$ fulfil the dispersion relation

$$\left[1 - \mathrm{i} \frac{\bar{\eta} \omega_1}{\varrho_0 a_0^2}\right] k^2 = \frac{\omega_1^2}{a_0^2}. \tag{4.13}$$

For a given circular frequency $\omega_1 > 0$ two corresponding wavenumbers,

$$k_{1,2} = \pm \frac{\omega_1}{a_0} \sqrt{\frac{1 + \mathrm{i}(\bar{\eta}\omega_1/\varrho_0 a_0^2)}{1 + (\bar{\eta}\omega_1/\varrho_0 a_0^2)^2}}, \tag{4.14}$$

result for waves propagating in the forward/backward direction. Via the nonlinear dependence on $\omega_1$ a slight dispersion is indicated and according to $\mathrm{Im}\,k_1 > 0$ and $\mathrm{Im}\,k_2 < 0$ the damping of the waves in the propagation direction becomes apparent. For $\bar{\eta}\omega_1/\varrho_0 a_0^2 \ll 1$ a Taylor expansion of the above dispersion relation gives

$$k_{1,2} = \pm \frac{\omega_1}{a_0} \left(1 + \mathrm{i} \frac{\bar{\eta}\omega_1}{2\varrho_0 a_0^2}\right), \tag{4.15}$$

leading to the fundamental solutions

$$P = \hat{P} \exp\left(\mp \frac{\bar{\eta}\omega_1^2}{2\varrho_0 a_0^3} x\right) \exp\left(\mathrm{i}\left[\pm \frac{\omega_1}{a_0} x - \omega_1 t\right]\right), \tag{4.16}$$

of damped harmonic waves with frequency-dependent damping coefficient $\bar{\eta}\omega_1^2/2\varrho_0 a_0^3$.

## 4.4. Solution of the non-equilibrium equations

Assuming again wave-like solutions of the harmonic form $P = \hat{P} \exp(\mathrm{i}[kx - \omega_1 t])$ of the PDE (4.11) for finite $\omega_0$, one obtains

$$\left[\left(1 - \mathrm{i} \frac{\bar{\eta}\omega_1}{\varrho_0 a_0^2} + S(-\omega_0 t) \frac{\bar{\eta}\omega_1^2}{\varrho_0 \omega_0 a_0^2}\right) k^2 - \frac{\omega_1^2}{a_0^2}\right] \hat{P} \exp(\mathrm{i}[kx - \omega_1 t]) = 0 \tag{4.17}$$

showing that the harmonic form does not solve the PDE (4.11). Thus, in the non-equilibrium case the waveform must be slightly inharmonic. However, in appendix B the error is estimated: it depends essentially on the square of the ratio $\omega_1/\omega_0$ of the circular wave frequency to the relaxation rate. As a consequence, the classical solution (4.16) based on the dispersion relation (4.13) applies for frequencies $\omega_1 \ll \omega_0$ to a good approximation.

# 5. Conclusion and outlook

Using a general analysis, common features of the discontinuous Lagrangian approach and the stochastic variational description have been discovered despite the different character (deterministic vs statistical) of both. The discontinuous Lagrangian approach is successfully generalized toward compressible flow, with damped acoustic waves considered as an example revealing the criterion that deviations from the classical theory are of order $\omega_1^2/\omega_0^2$, i.e. they may occur at very high wave frequencies $\omega_1$ if approaching the thermal relaxation rate $\omega_0$. Opening up the possibility for validating the discontinuous Lagrangian approach via experiment with parameter identification of the thermal relaxation rate in future.

As mentioned in §2.3, the question arises whether the discontinuous Lagrangian approach with spatial averaging instead of time averaging would deliver equivalent results in accordance with the

ergodic hypothesis and well known from turbulence theory. This question will be investigated next in order to gain more trust in this new approach.

Various generalizations of the discontinuous Lagrangian approach appear promising: having generalized the method from incompressible to compressible flow, the application to non-Newtonian fluids should be possible by replacing the dissipation rate for a Newtonian fluid in the Lagrangian (3.1) with the respective dissipation rate for a non-Newtonian one.

The third approach addressed here, the first integral approach, seems to be fully independent of the discontinuous Lagrangian approach and the stochastic variational description. At present the Lagrangian it delivers is restricted to steady flow. It nevertheless will form the subject of forthcoming research projects to find connections to the aforementioned approaches.

Apart from the three approaches discussed here, there are various other strategies that can be used to establish variational formulations in continuum mechanics, following again different ideas. One of them is the use of *nonlocal* functionals. As an example consider the variational principle reported in [40,41] for the convection-conduction equation.

Ethics. No ethical issues are raised by this contribution to fluid mechanics.

Data accessibility. This is a pure theoretical work without experimental or numerical data.

Authors' contributions. M.S. proposed the discontinuous Lagrangian for viscous flow and worked out the time-averaged equations of motion, F.M. did the the literature research on stochastic variational description and P.H.G. discussed and interpreted the theoretical results and drafted the manuscript. D.H. and M.S. elaborated the general compressible theory and the sound wave example. All authors gave final approval for publication.

Competing interests. We declare we have no competing interests.

Funding. F.M. acknowledges the financial support from the Thomas Gessmann-Stiftung for his doctoral project. M.S. and F.M. acknowledge the support from the German Research Foundation (DFG), SCHO 767/6-3.

Acknowledgements. We thank Pedro Jordan and Ana Bela Cruzeiro for valuable and very constructive hints. In addition P.H.G. is grateful to Durham University for the granting of an extended period of research leave, making this collaborative research possible.

# Appendix A. Equations of motion related to the Lagrangian (3.1)

The Euler–Lagrange equations (3.4)–(3.9) are a first integral of the equations of motion, different from the one reported in [24], but with the common feature that the equations of motion are obtained from their spatial and temporal derivatives as follows. Consider the useful identity

$$\nabla \cdot \left( i \ln \sqrt{\frac{\bar{\chi}}{\chi}} [2\eta \underline{D} + \eta' \nabla \cdot \boldsymbol{u} \, \underline{1}] \right)$$

$$= \underbrace{[2\eta \underline{D} + \eta' \nabla \cdot \boldsymbol{u} \, \underline{1}]}_{=:\underline{T}_v} \text{Im} \frac{\nabla \chi}{\chi} + i \ln \sqrt{\frac{\bar{\chi}}{\chi}} \underbrace{[\eta \Delta \boldsymbol{u} + (\eta + \eta') \nabla (\nabla \cdot \boldsymbol{u})]}_{=:f_v}$$

in which the viscous stress tensor $\underline{T}_v$ and the density $f_v$ of the viscous forces appear for the first time and derive from (3.9) the following evolution equation for the phase of the thermal excitation:

$$i D_t \ln \sqrt{\frac{\bar{\chi}}{\chi}} = \text{Im} \left[ \frac{D_t \chi}{\chi} \right] = \text{Im} \left( \frac{1}{2\varrho\bar{\chi}\chi} \left[ \eta \, \text{tr} \, \underline{D}^2 + \frac{\eta'}{2} (\nabla \cdot \boldsymbol{u})^2 \right] - i\omega_0 \frac{c_0 T}{\bar{\chi}\chi} \right)$$

$$= -\omega_0 T \exp \left( -\frac{s}{c_0} \right) =: -\omega, \tag{A 1}$$

with a modified thermal relaxation rate $\omega = \omega(s, \varrho) = \omega_0 T(s, \varrho) \exp(-s/c_0)$ depending very weakly on

the thermodynamic state. Furthermore,

$$
D_t \left[ \nabla \Phi + \alpha_i \nabla \beta_i + \operatorname{Im} \frac{\bar{\chi} \nabla \chi}{\omega_0} \right]
$$

$$
= \nabla \left[ D_t \Phi + \alpha_i D_t \beta_i + \operatorname{Im} \frac{\bar{\chi} D_t \chi}{\omega_0} \right] - \nabla \otimes \boldsymbol{u} \left[ \nabla \Phi + \alpha_i \nabla \beta_i + \operatorname{Im} \frac{\bar{\chi} \nabla \chi}{\omega_0} \right]
$$

$$
+ D_t \alpha_i \nabla \beta_i - D_t \beta_i \nabla \alpha_i - \frac{2}{\omega_0} \operatorname{Im}(D_t \chi \nabla \bar{\chi})
$$

$$
= \nabla \left[ \frac{u^2}{2} - e - \varrho \frac{\partial e}{\partial \varrho} - V \right] - \frac{\nabla \otimes \boldsymbol{u}}{\varrho \omega_0} \nabla \cdot \left( \mathrm{i} \ln \sqrt{\frac{\bar{\chi}}{\chi}} [2 \eta \underline{D} + \eta' \nabla \cdot \boldsymbol{u} \, \underline{1}] \right)
$$

$$
- (\nabla \otimes \boldsymbol{u}) \boldsymbol{u} - \operatorname{Im} \left( \frac{\nabla \bar{\chi}}{\varrho \omega_0 \bar{\chi}} \left[ \eta \operatorname{tr} \underline{D}^2 + \frac{\eta'}{2} (\nabla \cdot \boldsymbol{u})^2 \right] - 2\mathrm{i} c_0 T \frac{\nabla \bar{\chi}}{\bar{\chi}} \right)
$$

$$
= -\nabla \underbrace{\left[ e + \varrho \frac{\partial e}{\partial \varrho} \right]}_{h} - \nabla V - \frac{\nabla \otimes \boldsymbol{u}}{\varrho \omega_0} \left[ \underline{T}_{\mathrm{v}} \operatorname{Im} \frac{\nabla \chi}{\chi} + \mathrm{i} \ln \sqrt{\frac{\bar{\chi}}{\chi}} f_{\mathrm{v}} \right]
$$

$$
- \frac{1}{\varrho \omega_0} \left[ \eta \operatorname{tr} \underline{D}^2 + \frac{\eta'}{2} (\nabla \cdot \boldsymbol{u})^2 \right] \operatorname{Im} \frac{\nabla \bar{\chi}}{\bar{\chi}} + T \overbrace{2 c_0 \operatorname{Re} \frac{\nabla \bar{\chi}}{\bar{\chi}}}^{\nabla s},
$$

is obtained from the Euler–Lagrange equations (3.4–3.9); $h$ is the specific enthalpy given by $h = e - p/\varrho$. Considering all of the above, the material acceleration can be obtained from (3.7) as

$$
D_t \boldsymbol{u} = D_t \left[ \nabla \Phi + \alpha_i \nabla \beta_i + \frac{1}{\omega_0} \operatorname{Im}(\bar{\chi} \nabla \chi) \right] - D_t \left( \frac{1}{\varrho \omega_0} \left[ \underline{T}_{\mathrm{v}} \operatorname{Im} \frac{\nabla \chi}{\chi} + \mathrm{i} \ln \sqrt{\frac{\bar{\chi}}{\chi}} f_{\mathrm{v}} \right] \right)
$$

$$
= -\nabla h - \nabla V - \frac{\nabla \otimes \boldsymbol{u}}{\varrho \omega_0} \left[ \underline{T}_{\mathrm{v}} \operatorname{Im} \frac{\nabla \chi}{\chi} + \mathrm{i} \ln \sqrt{\frac{\bar{\chi}}{\chi}} f_{\mathrm{v}} \right]
$$

$$
- \frac{1}{\varrho \omega_0} \left[ \eta \operatorname{tr} \underline{D}^2 + \frac{\eta'}{2} (\nabla \cdot \boldsymbol{u})^2 \right] \operatorname{Im} \frac{\nabla \bar{\chi}}{\bar{\chi}}
$$

$$
- D_t \left( \frac{T_{\mathrm{v}}}{\varrho \omega_0} \operatorname{Im} \frac{\nabla \chi}{\chi} \right) - \mathrm{i} \ln \sqrt{\frac{\bar{\chi}}{\chi}} D_t \left( \frac{f_{\mathrm{v}}}{\varrho \omega_0} \right) - \mathrm{i} \underbrace{D_t \ln \sqrt{\frac{\bar{\chi}}{\chi}}}_{-\omega} \frac{f_{\mathrm{v}}}{\varrho \omega_0} + T \nabla s.
$$

Note that for constant specific heat and vanishing thermal expansion coefficient $s = c_0 \ln(T/T_0)$ and therefore $\omega = \omega_0$. In the case of more general thermodynamic constitutive equations $\omega \approx \omega_0$ is assumed. Finally, keeping the identity

$$
\nabla h - T \nabla s = \nabla \left[ e + \varrho \frac{\partial e}{\partial \varrho} \right] - \frac{\partial e}{\partial s} \nabla s = \frac{1}{\varrho} \nabla \left[ \varrho^2 \frac{\partial e}{\partial \varrho} \right] = \frac{\nabla p}{\varrho}
$$

in mind, an equation of motion of the form

$$
D_t \boldsymbol{u} = -\frac{\nabla p}{\varrho} - \nabla V + \frac{\eta}{\varrho} \Delta \boldsymbol{u} + \left( \eta + \frac{\eta'}{\varrho} \right) \nabla (\nabla \cdot \boldsymbol{u}) + \frac{1}{\varrho} \boldsymbol{f}_{\mathrm{n.e.}} \tag{A 2}
$$

is obtained, which is the Navier–Stokes–Duhem equation for constant viscosities plus additional fluctuating forces,

$$
\frac{1}{\varrho} \boldsymbol{f}_{\mathrm{n.e.}} := \frac{1}{\omega_0} \left\{ \left[ \frac{\eta}{\varrho} \operatorname{tr} \underline{D}^2 + \frac{\eta'}{2\varrho} (\nabla \cdot \boldsymbol{u})^2 \right] \operatorname{Im} \frac{\nabla \chi}{\chi} - \mathrm{i} \ln \sqrt{\frac{\bar{\chi}}{\chi}} \{ D_t + \nabla \otimes \boldsymbol{u} \} \left( \frac{f_{\mathrm{v}}}{\varrho} \right) - \{ D_t + \nabla \otimes \boldsymbol{u} \} \left( \frac{T_{\mathrm{v}}}{\varrho} \operatorname{Im} \frac{\nabla \chi}{\chi} \right) \right\}, \tag{A 3}
$$

as in the case of compressible flow.

# Appendix B. Time averaging of equation (4.17) and error estimation

By applying the time averaging operation (2.15) to Eq. (4.17), followed by division by $\hat{P} \exp(ikx)$, the relation

$$\left[1 - i\frac{\bar{\eta}\omega_1}{\varrho_0 a_0^2} + \frac{\langle S(-\omega_0 t)\exp(-i\omega_1 t)\rangle}{\langle\exp(-i\omega_1 t)\rangle}\frac{\bar{\eta}\omega_1^2}{\varrho_0\omega_0 a_0^2}\right]k^2 - \frac{\omega_1^2}{a_0^2} = 0 \tag{B1}$$

is obtained. It is subsequently shown that the time dependence becoming manifest in the factor $\langle S(-\omega_0 t)\exp(-i\omega_1 t)\rangle/\langle\exp(-i\omega_1 t)\rangle$ can be fully eliminated by a proper choice of the time interval $\tau$. First compute

$$\langle\exp(-i\omega_1 t)\rangle = \frac{1}{2\tau}\int_{t-\tau}^{t+\tau}\exp(-i\omega_1 t')dt' = -\frac{\exp(-i\omega_1 t')}{2i\tau\omega_1}\Big|_{t-\tau}^{t+\tau}$$
$$= \frac{\sin(\omega_1\tau)}{\omega_1\tau}\exp(-i\omega_1 t). \tag{B2}$$

For calculating $\langle S(-\omega_0 t)\exp(-i\omega_1 t)\rangle$, the sawtooth function $S(x)$ is expressed by its Fourier series,

$$S(x) = -2\sum_{n=1}^{\infty}\frac{(-1)^n}{n}\sin(nx),$$

implying

$$\langle S(-\omega_0 t)\exp(-i\omega_1 t)\rangle = -\frac{1}{i}\sum_{n=1}^{\infty}\frac{(-1)^n}{n}\langle 2i\sin(-n\omega_0 t)\exp(-i\omega_1 t)\rangle$$
$$= i\sum_{n=1}^{\infty}\frac{(-1)^n}{n}[\langle\exp(-i[\omega_1 + n\omega_0]t)\rangle - \langle\exp(-i[\omega_1 - n\omega_0]t)\rangle]$$
$$= i\sum_{n=1}^{\infty}\frac{(-1)^n}{n}\left[\frac{\sin([\omega_1 + n\omega_0]\tau)}{[\omega_1 + n\omega_0]\tau}\exp(-i[\omega_1 + n\omega_0]t)\right.$$
$$\left. - \frac{\sin([\omega_1 - n\omega_0]\tau)}{[\omega_1 - n\omega_0]\tau}\exp(-i[\omega_1 - n\omega_0]t)\right].$$

For convenience the time interval $\tau$ according to $\omega_0\tau = 2\pi N$ with integer number $N$, is chosen simplifying the above formula to

$$\langle S(-\omega_0 t)\exp(-i\omega_1 t)\rangle$$
$$= i\sin(\omega_1\tau)\exp(-i\omega_1 t)\sum_{n=1}^{\infty}\frac{(-1)^n}{n}\left[\frac{\exp(-in\omega_0 t)}{nN + \omega_1\tau} + \frac{\exp(in\omega_0 t)}{nN - \omega_1\tau}\right]. \tag{B3}$$

Finally, the following is obtained:

$$\frac{\langle S(-\omega_0 t)\exp(-i\omega_1 t)\rangle}{\langle\exp(-i\omega_1 t)\rangle} = i\omega_1\frac{\tau}{N}\sum_{n=1}^{\infty}\frac{(-1)^n}{n^2}\left[\frac{\exp(-in\omega_0 t)}{1 + \frac{\omega_1\tau}{nN}} + \frac{\exp(in\omega_0 t)}{1 - \frac{\omega_1\tau}{nN}}\right]. \tag{B4}$$

By considering $\tau/N = 2\pi/\omega_0$, $\omega_1\tau/(nN) \ll 1$ and truncating the sum after the leading Fourier order, the error in the dispersion relation (B 1) can be estimated as

$$\frac{\langle S(-\omega_0 t)\exp(-i\omega_1 t)\rangle}{\langle\exp(-i\omega_1 t)\rangle}\frac{\bar{\eta}\omega_1^2}{\varrho_0\omega_0 a_0^2} \approx -i\left(\frac{\omega_1}{\omega_0}\right)^2\frac{2\pi\bar{\eta}\omega_1}{\varrho_0 a_0^2}\cos(\omega_0 t), \tag{B5}$$

where the square of the ratio $\omega_1/\omega_0$ of the circular wave frequency and the relaxation rate is decisive, i.e. the classical solution (4.16) based on the dispersion relation (4.13) applies for frequencies $\omega_1 \ll \omega_0$.

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
