## [Reviewer comments · Royal Society Open Science]

Review History

Decision letter (RSOS-171620.R0)

23-Nov-2017

Dear Dr Scholle:

Manuscript ID RSOS-171620 entitled "Lagrangians for viscous flow: deterministic versus stochastic approaches" which you submitted to Royal Society Open Science, has been assessed. The comments from editor are included at the bottom of this letter.

In view of the criticisms of the Associate Editor, the manuscript has been rejected in its current form. However, a new manuscript may be submitted which takes into consideration these comments.

Please note that resubmitting your manuscript does not guarantee eventual acceptance, and that your resubmission will be subject to peer review before a decision is made.

Your resubmitted manuscript should be submitted by 23-May-2018. If you are unable to submit by this date please contact the Editorial Office.

Please note that Royal Society Open Science will introduce article processing charges for all new submissions received from 1 January 2018. Charges will also apply to papers transferred to Royal Society Open Science from other Royal Society Publishing journals, as well as papers submitted as part of our collaboration with the Royal Society of Chemistry (<http://rsos.royalsocietypublishing.org/chemistry>). If your manuscript is submitted and accepted for publication after 1 Jan 2018, you will be asked to pay the article processing charge, unless you request a waiver and this is approved by Royal Society Publishing. You can find out more about the charges at <http://rsos.royalsocietypublishing.org/page/charges>. Should you have any queries, please contact openscience@royalsociety.org.

on behalf of Professor Alban Potherat (Associate Editor) and Miles Padgett (Subject Editor)
openscience@royalsociety.org

Associate Editor Comments to Author:

Dear Dr Scholle

Each article must be self-contained. Therefore, referencing other paper does not dispense the authors from writing a comprehensive introduction, defining all variables and clearly stating their conclusions.

Your manuscript is rejected on these grounds but you may want to write a full article based on it.

your sincerely
Alban Potherat

Author's Response to Decision Letter for (RSOS-171620.R0)

Dear Mrs Power,

The revised form of our manuscript is now self-contained and starts with a comprehensive introduction. Definitions of all variables are provided and the conclusions are more clearly stated.

RSOS-172136.R0

Review form: Reviewer 1

Is the manuscript scientifically sound in its present form?

Yes

Are the interpretations and conclusions justified by the results?

Yes

Is the language acceptable?

Yes

Is it clear how to access all supporting data?

Not Applicable

Do you have any ethical concerns with this paper?

No

Have you any concerns about statistical analyses in this paper?

No

Recommendation?

Major revision is needed (please make suggestions in comments)

Comments to the Author(s)

The authors obtain an extended Navier-Stokes (NS) type equation, which reduces to the classical NS eq. in the appropriate limit, using a variational formalism and a proposed Lagrangian density. This is an approach that I found very interesting. However this was introduced by two of the authors in an article from 2017, in the same journal (Ref. 12).

They then show that the additional non-classical contributions vanish when time averaged, in the assumption that the phases of the thermal fluctuation vary much faster in time compared to the characteristic timescales of the flow. This is the only novel result presented, as far as I can see. Expanding on the validity of this assumption from the physical perspective of spatial scales would strengthen this work, as splitting eq. 2.7 just into fast and slow terms leads to the recovery of the classical NS in a seemingly trivial way. Basically, why do I need eq 2.7 in the first place if the flow (as in all scales up to the Kolmogorov scale) is always much slower than the phases of the thermal fluctuations? It would be interesting to list the spatial scale (compared to the Kolmogorov scale) at which the timescale of the non-equilibrium restoration force is order unity to the other terms.

The comparison with stochastic variational description is welcomed, but it's quite light.

While I enjoyed reading this article, I find it to be more of a comment on their previous work than a standalone research article. The authors should greatly expand on the "deterministic versus stochastic approaches" aspect of the work. At this point, I do not recommend publication of this work.

Review form: Reviewer 2

Is the manuscript scientifically sound in its present form?

No

Are the interpretations and conclusions justified by the results?

No

Is the language acceptable?

Yes

Is it clear how to access all supporting data?

Not Applicable

Do you have any ethical concerns with this paper?

No

Have you any concerns about statistical analyses in this paper?

No

Recommendation?

Major revision is needed (please make suggestions in comments)

Comments to the Author(s)

Attached (see Appendix A).

Decision letter (RSOS-172136.R0)

22-Jan-2018

Dear Dr Scholle:

Manuscript ID RSOS-172136 entitled "Lagrangians for viscous flow: deterministic versus stochastic approaches" which you submitted to Royal Society Open Science, has been reviewed. The comments from reviewer(s) are included at the bottom of this letter.

In view of the criticisms of the reviewer(s), I must decline the manuscript for publication in Royal

Society Open Science at this time. However, a new manuscript may be submitted which takes into consideration these comments.

Please note that resubmitting your manuscript does not guarantee eventual acceptance, and that your resubmission will be subject to re-review by the reviewer(s) before a decision is rendered.

You will be unable to make your revisions on the originally submitted version of your manuscript. Instead, revise your manuscript using a word processing program and save it on your computer.

You may also click the below link to start the resubmission process (or continue the process if you have already started your resubmission) for your manuscript. If you use the below link you will not be required to login to ScholarOne Manuscripts.

*** PLEASE NOTE: This is a two-step process. After clicking on the link, you will be directed to a webpage to confirm. ***

https://mc.manuscriptcentral.com/rsos?URL_MASK=77d6cdbfd57648d0a31c571bdb02a641

Because we are trying to facilitate timely publication of manuscripts submitted to Royal Society Open Science, your resubmitted manuscript should be submitted by 22-Jul-2018. If you are unable to submit by this date please contact the Editorial Office for options.

Please note that Royal Society Open Science will introduce article processing charges for all new submissions received from 1 January 2018. Charges will also apply to papers transferred to Royal Society Open Science from other Royal Society Publishing journals, as well as papers submitted as part of our collaboration with the Royal Society of Chemistry (<http://rsos.royalsocietypublishing.org/chemistry>). If your manuscript is submitted and accepted for publication after 1 Jan 2018, you will be asked to pay the article processing charge, unless you request a waiver and this is approved by Royal Society Publishing. You can find out more about the charges at <http://rsos.royalsocietypublishing.org/page/charges>. Should you have any queries, please contact openscience@royalsociety.org.

I look forward to a resubmission.

on behalf of Professor Alban Potherat (Associate Editor) and Miles Padgett (Subject Editor)
openscience@royalsociety.org

Associate Editor Comments to Author (Professor Alban Potherat):

I have now heard from the reviewers. Both indicate that whilst there is merit in the ideas

presented in this paper, the material at hand would require a complete rework to warrant publication. Both referees give indication that the authors may want to consider if submitting a further paper based on their ideas.

Reviewer comments to Author:

Reviewer: 1

Comments to the Author(s)

The authors obtain an extended Navier-Stokes (NS) type equation, which reduces to the classical NS eq. in the appropriate limit, using a variational formalism and a proposed Lagrangian density. This is an approach that I found very interesting. However this was introduced by two of the authors in an article from 2017, in the same journal (Ref. 12).

They then show that the additional non-classical contributions vanish when time averaged, in the assumption that the phases of the thermal fluctuation vary much faster in time compared to the characteristic timescales of the flow. This is the only novel result presented, as far as I can see. Expanding on the validity of this assumption from the physical perspective of spatial scales would strengthen this work, as splitting eq. 2.7 just into fast and slow terms leads to the recovery of the classical NS in a seemingly trivial way. Basically, why do I need eq 2.7 in the first place if the flow (as in all scales up to the Kolmogorov scale) is always much slower than the phases of the thermal fluctuations? It would be interesting to list the spatial scale (compared to the Kolmogorov scale) at which the timescale of the non-equilibrium restoration force is order unity to the other terms.

The comparison with stochastic variational description is welcomed, but it's quite light.

While I enjoyed reading this article, I find it to be more of a comment on their previous work than a standalone research article. The authors should greatly expand on the "deterministic versus stochastic approaches" aspect of the work. At this point, I do not recommend publication of this work.

Reviewer: 2

Comments to the Author(s)

Attached.

Author's Response to Decision Letter for (RSOS-172136.R0)

See Appendix B.

RSOS-181595.R0

Review form: Reviewer 1

Is the manuscript scientifically sound in its present form?

Yes

Are the interpretations and conclusions justified by the results?

Yes

Is the language acceptable?

Yes

Is it clear how to access all supporting data?

Not Applicable

Do you have any ethical concerns with this paper?

No

Have you any concerns about statistical analyses in this paper?

No

Recommendation?

Accept as is

Comments to the Author(s)

I have read the new version of the article and I have no detailed comments to make.

The article is now in a state that can stand on its own (this has nothing to do with the number of pages, just with the work presented in those pages) and I consider it's worth publication in its current form.

The impact of the non-equilibrium terms for different flow regimes may yield novel results or give trivial answers. The community at large will ultimately decide on the practical relevance of the approach. However, from a mathematical perspective, I fully agree with the authors on their point:

"... the question is not if this equation is needed or not, but the question is if an alternative Lagrangian exists that provides an equation of motion without additional non-equilibrium terms."

Decision letter (RSOS-181595.R0)

07-Dec-2018

Dear Dr Scholle,

I am pleased to inform you that your manuscript entitled "Competing Lagrangians for incompressible and compressible viscous flow" is now accepted for publication in Royal Society Open Science.

Royal Society Open Science operates under a continuous publication model (<http://bit.ly/cpFAQ>). Your article will be published straight into the next open issue and this

will be the final version of the paper. As such, it can be cited immediately by other researchers. As the issue version of your paper will be the only version to be published I would advise you to check your proofs thoroughly as changes cannot be made once the paper is published.

You have the opportunity to archive your accepted, unbranded manuscript, but access to the full text must be embargoed until publication.

Articles are normally press released. For this to be effective we set an embargo on news coverage corresponding to the publication date of the article. We request that news media and the authors do not publish stories ahead of this embargo (when final version of the article is available).

on behalf of Professor Alban Potherat (Associate Editor) and Miles Padgett (Subject Editor)
openscience@royalsociety.org

Reviewer comments to Author:
Reviewer: 1

Comments to the Author(s)
I have read the new version of the article and I have no detailed comments to make.

The article is now in a state that can stand on its own (this has nothing to do with the number of pages, just with the work presented in those pages) and I consider it's worth publication in its current form.

The impact of the non-equilibrium terms for different flow regimes may yield novel results or give trivial answers. The community at large will ultimately decide on the practical relevance of the approach. However, from a mathematical perspective, I fully agree with the authors on their point:

"... the question is not if this equation is needed or not, but the question is if an alternative Lagrangian exists that provides an equation of motion without additional non-equilibrium terms."

Appendix A

The goal of the set of equation that this group investigates is to modify the usual incompressible viscous Navier-Stokes equations so that there is a second reservoir for the dissipated energy, is not strictly dissipative and might have a Hamiltonian formulation. They use Clebsch variables to represent where the energy removed by viscous effects goes and in this paper, they have added stochastic effects.

What is missing from the beginning of this paper is a physical motivation for introducing these equations, which seem to be a variation upon, or parameterisation of, the Navier-Stokes-Duhem equations, without any reference to what those are. From their earlier paper, they are in a contribution by Olsson that I don't have ready access to. Also from their earlier paper, the introduction of Clebsch variables might be relevant to a fluid with dislocations.

As far as this being an extension of true incompressible viscous Navier-Stokes equation, I would like to believe that an extension with a Stokes viscosity and a Hamiltonian formulation exists where the loss of kinetic energy through the viscous terms can be replaced by suitable additional physics.

Other than going back to the compressible Navier-Stokes equation, I don't believe there is and don't believe that adding a Clebsch pair, inspired by the Madelung transformation for quantum media, which can generate a phase velocity and even a pseudovorticity in the manner of Berry, plus some stochasticity, is the solution.

One fundamental problem with this approach is the question of helicity. What role helicity plays in turbulent flows has never been adequately addressed, except it is there in observed vortical structures and cannot be represented by a single Clebsch pair. Due to the extra potential needed to ensure incompressibility, the integrated helicity for a single Clebsch pair is identically zero. Which means for a physical fluid, multiple Clebsch pairs will be needed.

Furthermore, there is some literature on the using Clebsch pairs for diagnosis which the authors do not seem to know. For example

Study on the Eulerian–Lagrangian analysis of Navier–Stokes turbulence, K. Ohkitani and P. Constantin, *Physics of Fluids* 20, 075102 (2008);

C. Cartes, M. Bustamante, and M. E. Brachet, “Generalized Eulerian–Lagrangian description of Navier–Stokes dynamics,” *Phys. Fluids* 19, 077101 (2007).

and most recently:

K. Ohkitani, Study of the 3D Euler equations using Clebsch potentials: dual mechanisms for geometric depletion, *Nonlinearity* 31, R25 (2018).

Koji Ohkitani

Published 2 January 2018

If they pose their solutions as solutions for a Duhem liquid, which I have never hear of unless it is something like Silly Putty, I will listen. However, as currently posed, I am against publication.

Appendix B

Authors' note to the Editor

We are grateful to the editor for providing us with the opportunity to revise and resubmit our manuscript and thank the two referees for their very constructive and insightful comments, enabling us to carry out a complete revision of the work, both rigorous and thorough, leading to a far superior article via which the essence of our work is more clearly articulated.

We are confident, based on the thorough revision carried out, that the new version of the manuscript addresses the concerns raised by the referees in full, showing the connection to existing approaches/theories.

Authors' reply to the Referees

We would like to begin by thanking the referee for their frank and honest opinion as to the content of our manuscript and for taking the time to point us in the right direction as to its essential improvement.

Reviewer: 1

The authors obtain an extended Navier-Stokes (NS) type equation, which reduces to the classical NS eq. in the appropriate limit, using a variational formalism and a proposed Lagrangian density. This is an approach that I found very interesting. However this was introduced by two of the authors in an article from 2017, in the same journal (Ref. 12).

Reviewer's comment: They then show that the additional non-classical contributions vanish when time averaged, in the assumption that the phases of the thermal fluctuation vary much faster in time compared to the characteristic timescales of the flow. This is the only novel result presented, as far as I can see. Expanding on the validity of this assumption from the physical perspective of spatial scales would strengthen this work, as splitting eq. 2.7 just into fast and slow terms leads to the recovery of the classical NS in a seemingly trivial way.

Authors' reply: We thank the reviewer for these fruitful hints. We agree that spatial averaging would provide an interesting alternative to time averaging and would strengthen this work, while expecting the same result as in turbulence theory due to ergodic hypothesis. We started investigating in this variant but need more time to work it out. The main reason for this is that the discontinuities appearing in the theory become manifest as wave-like structures on a microscopic scale. By spatial averaging they become micro-solitons the detailed analysis of which is required before serious conclusions about their physical meaning can be drawn. The necessary studies of the solutions of the corresponding evolution equation for the thermal excitation will require some time, but we are confident to provide results in forthcoming papers. At least

we have discussed the idea of spatial averaging in Sec. 2(c) and in the outlook section 5.

Reviewer’s comment: Basically, why do I need eq 2.7 in the first place if the flow (as in all scales up to the Kolmogorov scale) is always much slower than the phases of the thermal fluctuations? It would be interesting to list the spatial scale (compared to the Kolmogorov scale) at which the timescale of the non-equilibrium restoration force is order unity to the other terms. The comparison with stochastic variational description is welcomed, but it’s quite light.

Authors’ reply: The equation of motion 2.9 (2.7 in the original manuscript) results from the Lagrangian 2.1 by variation, so the question is not if this equation is needed or not, but the question is if an alternative Lagrangian exists that provides an equation of motion without additional non-equilibrium terms. Based on prior works, from Millikan to our RSOS paper from 2017, no alternative Lagrangian delivering full Navier-Stokes equations (without introducing additional fields without physical meaning in the sense of a weak formulation) is available, apart from the one provided in the framework of the first integral approach in Sec. 2(d).

However, the question arises if the non-equilibrium terms occurring in the equation of motion are relevant in turbulent considering the time Kolmogorov scale τ_η . At least, by $2\pi/\tau_\eta > \omega_0$ a rough criterium is given for a Reynolds number regime where the thermal fluctuations and the no-classical terms occurring in the equations of motion should be without any relevance. This opens the perspective to use the discontinuous Lagrangian approach directly without time averaging, making the implementation of it easier. Without any doubt it would have been more consequent and satisfying to list a separate spatial scale for comparison with the Kolmogorov scale. Unfortunately an application for funding a research project related to the application of our unconventional methods to turbulent flow has been refused by the DFG, so our research capacities are quite restricted and we are sorry that we can address your suggestion only in this minimalistic manner. At least our rough considerations has been added in a separate section 2(e), where also the use of the alternative approaches is discussed at different Re-regimes. We are thankful to the reviewer pointing on this relevant question.

Reviewer’s comment: While I enjoyed reading this articulated, I find it to be more of a comment on their previous work than a standalone research article. The authors should greatly expand on the ”deterministic versus stochastic approaches” aspect of the work. At this point, I do not recommend publication of this work.

Authors' reply

We agree that the original version of the article was at an early stage. Meanwhile we gained some progress on this field, especially by extending the theory from incompressible to compressible flow and working out an example which both have been added to the revised version. Since reviewer 2 had relevant concerns about the foundations of our approach in general and doubted its validity for compressible flow in particular, it was necessary to extend the approach to compressible flow. Although a Lagrangian is already provided in the prior paper (RSOS 2017), its associated Euler-Lagrange equations and resulting equations of motion have never been worked out and discussed, showing that the formalism equally well works for compressible as for incompressible flows. We also provide a slight modification of the Lagrangian leading to a simplified treatment. In order to demonstrate its capabilities, the method is applied to damped acoustic waves, which is elaborated in section 4. The treatment of damped acoustic waves includes also the use of statistics and provides a criterion up to which frequency the classical solution remains valid. Unfortunately no comparative results coming from stochastic variational approach are available at the moment, as A. B. Cruzeiro stated in a private communication. In the meantime (Apr. 2018) also an alternative approach has been published, the first integral approach, providing a continuous Lagrangian but being restricted to steady and therefore laminar flows. We also address this approach.

We hope that the reviewer agree that the revised version of our manuscript, which has twice the number of pages as the original one, can be considered as a standalone research article rather than a comment to our prior work now.

Reviewer: 2

Reviewer's comment: The goal of the set of equation that this group investigates is to modify the usual incompressible viscous Navier-Stokes equations so that there is a second reservoir for the dissipated energy, is not strictly dissipative and might have a Hamiltonian formulation.

Authors' reply: This is indeed not the goal: it was e.g. the goal of the prior paper, RSOS 2017; 4(2), and has been reached considering thermal degrees of freedom. In fact the goal is to show how the prior work can be reinterpreted from a statistical viewpoint, ending up with a modified approach, which we now call a 'pseudo-statistical' one, that is closer to the classical theory. We regret that the real intention of our manuscript was misunderstood and thank the reviewer for pointing to this. To make it clearer, we have revised the abstract and Sect. 1(d) (formerly 1(c)), hoping that the aim of the paper is now much clearer and unambiguous.

Reviewer's comment: They use Clebsch variables to represent where the energy removed by viscous effects goes and in this paper, they have added stochastic effects.

Authors' reply: In fact we use Clebsch variables, but the dissipated energy does not go to the Clebsch variables, it goes to the internal energy (thermal energy, heat) in accordance with the first law of thermodynamics and causes an increase of the entropy in accordance with the second law of thermodynamics. This has been discussed in the prior paper, RSOS 2017; 4(2), but for clarification we added some sentences for explaining the energy transfer. We again thank the reviewer for advising us to this.

Reviewer's comment: What is missing from the beginning of this paper is a physical motivation for introducing these equations, which seem to be a variation upon, or parameterisation of, the Navier-Stokes-Duhem equations, without any reference to what those are. From their earlier paper, they are in a contribution by Olsson that I don't have ready access to. Also from their earlier paper, the introduction of Clebsch variables might be relevant to a fluid with dislocations.

Authors' reply: The Navier-Stokes-Duhem equations are simply the Navier-Stokes equations for compressible flow. Although this terminology is not widely spread, we prefer it for distinguishing the compressible from the incompressible case, since 'Navier-Stokes equations' is mostly understood as the Navier-Stokes equations for incompressible flow. We have added sub-sentence explaining the latter. Moreover, we added a reference to a second textbook by Belevich (next to Olsson's one) making use of this terminology.

Reviewer's comment: As far as this being an extension of true incompressible viscous Navier-Stokes equation, I would like to believe that an extension with a Stokes viscosity and a Hamiltonian formulation exists where the loss of kinetic energy through the viscous terms can be replaced by suitable additional physics. Other than going back to the compressible Navier-Stokes equation, I don't believe there is and don't believe that adding a Clebsch pair, inspired by the Madelung transformation for quantum media, which can generate a phase velocity and even a pseudovorticity in the manner of Berry, plus some stochasticity, is the solution.

Authors' reply: We are surprised about this statement since compressible flow has not been considered throughout the entire manuscript (original submitted version), therefore we do not understand this point of criticism. Nevertheless we address this point in the revised version by adding a new section in which the theory for compressible flow is worked out in detail, showing that the formalism equally well works for compressible as for incompressible flows. In order to demonstrate its capabilities, the method is applied to damped acoustic waves, which is elaborated in the following section. We again thank the reviewer for bringing our attention to this relevant question.

Reviewer's comment: One fundamental problem with this approach is the question of helicity. What role helicity plays in turbulent flows has never been

adequately addressed, except it is there in observed vortical structures and cannot be represented by a single Clebsch pair. Due to the extra potential needed to ensure incompressibility, the integrated helicity for a single Clebsch pair is identically zero. Which means for a physical fluid, multiple Clebsch pairs will be needed. Furthermore, there is some literature on the using Clebsch pairs for diagnosis which the authors do not seem to know. For example Study on the Eulerian–Lagrangian analysis of Navier–Stokes turbulence, K. Ohkitani and P. Constantin, *Physics of Fluids* 20, 075102 (2008); C. Cartes, M. Bustamante, and M. E. Brachet, “Generalized Eulerian–Lagrangian description of Navier–Stokes dynamics,” *Phys. Fluids* 19, 077101 (2007). and most recently: K. Ohkitani, Study of the 3D Euler equations using Clebsch potentials: dual mechanisms for geometric depletion, *Nonlinearity* 31, R25 (2018). Koji Ohkitani Published 2 January 2018.

Authors’ reply: We are aware of this issue and addressed this question already in our two earlier papers RSOS 2017; 4(2) and PLA 2016; 380(40). Being also aware that meanwhile decades of research has been spent to the question of the existence of the Clebsch representation depending on the vortex topology of the flow, we are thankful to the reviewer referencing the most recent articles published on this research field, which we have cited now. Nevertheless, it is not our aim to provide an own contribution to the latter, in fact the aim of this paper is a different one, as already mentioned above. To hedge one’s bets, we have replaced the single Clebsch pair $\alpha\nabla\beta$ by $\alpha_i\nabla\beta_i = \alpha_1\nabla\beta_1 + \alpha_2\nabla\beta_2 + \dots$ leaving it open how many pairs are needed for the individual flow topology.

Reviewer’s comment: If they pose their solutions as solutions for a Duhem liquid, which I have never hear of unless it is something like Silly Putty, I will listen.

Authors’ reply

We are again sorry for this misunderstanding: the solutions of the Navier-Stokes-Duhem equations (= Navier-Stokes equation for compressible flow) are not the solutions for a Duhem fluid. At the moment we prefer focussing our research to the Newtonian case with application to examples like the damped acoustic waves provided in Sec. 4. Nevertheless, there is indeed a perspective to apply the discontinuous Lagrangian approach to non-Newtonian frames, which is slightly sketched in the outlook section 5.